# Macrophage Activation Syndrome in Viral Sepsis

**DOI:** 10.3390/v16071004

**Published:** 2024-06-22

**Authors:** Despoina Papageorgiou, Charalambos Gogos, Karolina Akinosoglou

**Affiliations:** 1Department of Medicine, University of Patras, Rio, 26504 Patras, Greece; cgogos@med.upatras.gr (C.G.); akin@upatras.gr (K.A.); 2Metropolitan General Hospital, 15562 Athens, Greece; 3Department of Internal Medicine and Infectious Diseases, University of Patras, Rio, 26504 Patras, Greece

**Keywords:** macrophage activation syndrome, secondary hemophagocytic lymphohistiocytosis, viral infections, COVID-19

## Abstract

Macrophage activation syndrome (MAS) is a life-threatening systemic hyperinflammatory syndrome triggered by various infections, particularly viral infections, autoimmune disorders, and malignancy. The condition is characterized by an increased production of proinflammatory cytokines resulting in a cytokine storm and has been associated with poor clinical outcomes. During the COVID-19 pandemic, patients with severe manifestations developed features similar to those of MAS, although these characteristics remained well defined within the lung. Additionally, other viral infections including EBV, the herpes family of viruses, hepatitis viruses, influenza, HIV, and hemorrhagic fevers can be complicated by MAS. The diagnosis and management of the condition remain challenging due to the lack of consensus on specific guidelines, especially among the adult population. Currently, therapeutic options primarily rely on medications that are typically used to treat primary hemophagocytic lymphohistiocytosis, such as corticosteroids and etoposide. In addition, cytokine-targeted therapies present promising treatment options. The objective of this review is to discuss the emergence of MAS in the context of viral infections including, but not limited to, its occurrence in COVID-19.

## 1. Introduction

Macrophage activation syndrome (MAS) is a life-threatening systemic hyperinflammatory syndrome typically associated with autoimmune/autoinflammatory diseases. It is considered to be a form of secondary hemophagocytic lymphohistiocytosis (sHLH), which is triggered by various infections, autoimmune diseases and malignancy. In contrast, primary HLH (pHLH) occurs due to genetic disorders [1]. However, distinguishing between secondary HLH and primary HLH in clinical settings is challenging. MAS and HLH are sometimes used interchangeably or misused in studies, while distinguishing between triggers is also difficult.

The main pathophysiologic feature of MAS is excessive activation and expansion of T cells, particularly cytotoxic CD8+ T cells, and macrophages. This prolonged immune activation results in the increased production of proinflammatory cytokines, leading to a cytokine storm [2]. These cytokines include IFNγ, TNFα, IL-2, IL-1, IL-6, IL-18, as well as macrophage colony-stimulating factor (M-CSF) [3]. Although, the exact mechanism remains unknown, MAS shares many etiologic similarities with pHLH, which is a result of homozygous or compound heterozygous mutations in genes involved in the perforin-mediated pathway of cytolysis shared by NK cells and cytotoxic CD8 T cells. Interestingly, heterozygous mutations in known pHLH genes are increasingly recognized in MAS patients [4]. Clinical characteristics of MAS include non-remitting high fever, hepatomegaly/splenomegaly, lymphadenopathy, hemorrhagic manifestations, and multiorgan dysfunction, while laboratory findings typically involve cytopenias, hyperferritinemia, coagulopathy, abnormal liver function tests, hypertriglyceridemia, hypofibrinogenemia, and markers of immune activation. Hemophagocytosis is also commonly observed [5].

The diagnosis and management of MAS/sHLH pose significant challenges to clinicians due to the heterogenous nature of triggering factors and underlying conditions, the rarity of the condition, and the lack of consensus regarding broad classification [1]. The diagnosis primarily relies on the fulfillment of at least five out of eight clinical and laboratory criteria outlined in the HLH-2004 protocol [1]. However, these guidelines mostly apply to the diagnosis and treatment of pHLH in the pediatric population [6]. Alternatively, another effective diagnostic tool, the HScore, has been developed and is widely used to assess an individual’s risk for both pHLH and sHLH [7]. Of note, specific diagnostic guidelines for MAS in the setting of autoimmune diseases have been developed [8,9].

During the COVID-19 pandemic, a significant number of patients developed severe complications such as the virus-induced cytokine storm and MAS/HLH, which are associated with poor clinical outcomes, highlighting the substantial role of infectious triggers, particularly viral infections, in the pathogenesis of HLH [10]. The objective of this review is to investigate the occurrence of MAS/HLH within the framework of viral infections, including—but not limited to—its emergence in patients with COVID-19.

## 2. Materials and Methods

We performed a literature review, searching the Pubmed database from September 2003 until May 2024 for viral infections associated with MAS. We used the terms “Macrophage activation syndrome” OR “secondary hemophagocytic lymphohistiocytosis” AND the viral infection of interest, including “COVID-19”, “EBV”, “CMV”, “HSV-1”, “HSV-2”, “VZV”, “Influenza”, “H1N1” “Adenovirus”, “Parvovirus B19”, “HAV”, “HBV”, “HCV”, “HIV”, “Ebola virus”, and “dengue fever”. Relative publications were identified by their abstracts and title by two independent reviewers. Respective articles were also hand-searched for relative references. Only English articles were studied. Duplicates and non-human articles were excluded, as well as articles irrelevant to virus-associated MAS, such as papers inconclusive about MAS triggers or sHLH associated with etiology other than infection. Case reports were excluded. In the absence of relevant findings, and following extensive discussion, the authors decided that, for the sake of information completeness, case reports would be recorded in a table for parvovirus-B19 only. The results were discussed and disagreements resolved. A study flowchart is shown in Figure 1.

## 3. Discussion

### 3.1. MAS and COVID-19

Patients with SARS-CoV-2 infection may experience a range of clinical manifestations, from asymptomatic to severe disease and critical illness. Severe cases are characterized by a pronounced immune dysregulation with lymphopenia and increased expression of inflammatory mediators resulting in a cytokine storm. Early studies suggested that COVID-19 cytokine storms share significant similarities with MAS [11]. Specifically, interleukin-1β (IL-1β), IL-2, IL-6, IL-7, IL-17, tumor necrosis factor-α (TNF-α), granulocyte colony-stimulating factor (G-CSF), C-X-C motif chemokine ligand 10 (CXCL10), and C-C motif chemokine ligand 3 (CCL3) were reported to be upregulated in patients with severe COVID-19, especially among patients treated in intensive care units (ICUs) [12,13]. COVID-19 induces alternations in cytokine profiles that exhibit similarities to those seen in MAS. Cytokines such as IP10, IL-18, and macrophage colony-stimulating factor (M-CSF), which are key molecules in MAS, were identified as the most predictive of severe disease [14,15,16,17]. In addition, laboratory parameters of severely ill COVID-19 patients, such as cytopenia, hyperferritinemia, hyperfibrinogenemia, and hypoalbuminemia, resembled those of MAS, although they did not exhibit the typical features of the condition [18,19,20]. Of note, particularly among pediatric population with COVID-19, there is a subset of patients that may develop multisystem inflammatory syndrome (MIS-C), a rare post-infectious hyperinflammatory disorder [21]. The clinical and immunological features of MIS-C resemble those of MAS, complicating the distinction between these two conditions [22,23,24]. Additionally, patients diagnosed with MIS-C were found to carry heterozygous mutations in pHLH or HLH-associated genes [25]. Interestingly, several authors observed that patients with MIS-C may also meet the diagnostic criteria for MAS [26,27].

The overlapping profiles between severe COVID-19 and MAS have led to the hypothesis that the latter may be involved in COVID-19 pathogenesis and may drive some of its serious complications including ARDS [28]. Due to a temporary virus-induced immunodeficiency state which is associated with lymphopenia and NK cell abnormalities, the underlying mechanism of MAS in COVID-19 may resemble a pHLH-like phenotype. In contrast, a more typical MAS/sHLH presentation could be the result of an exaggerated immune response that clears the virus but induces pulmonary tissue damage and ARDS [28]. Of importance, the role of macrophages in severe COVID-19 cases is pivotal as the development of ARDS largely depends on a dysregulated macrophage activation pattern and the accumulation of activated macrophages in the lung [29,30]. Moreover, a post-mortem study identified diffuse interstitial pneumonia-like/MAS-like changes in patients with COVID-19 suggesting that these histopathological findings could represent a variant of MAS. Of interest, as these changes appear to be a late development of the disease, direct viral infection is unlikely to induce the macrophage dysregulation [31].

The immunological and pathological manifestations of MAS in COVID-19 patients are mainly observed to be lung-centric rather than systematic. Also, there are differences including the absence of organomegaly and the concentration of serum ferritin which is increased but lower than that in typical MAS [32,33]. In addition, the coagulation profile of COVID-19 patients differs from those with typical MAS, as indicated by the normal circulatory fibrinogen with elevated d-Dimers. Hence, these parameters may point towards pulmonary intravascular coagulopathy rather than diffuse intravascular coagulation (DIC) which is a feature of typical MAS [28,34].

The unique presentation of MAS in COVID-19 patients poses a great diagnostic challenge. Many studies have reported that a considerable number of patients do not meet the diagnostic criteria of HLH or the HScore [35,36,37]. In order to address this matter, studies focused on developing criteria which will provide a better characterization of COVID-19 cytokine storm in the context of the usage of immunomodulatory therapies [38,39]. Also, some studies have analyzed serum biomarker profiles to identify target molecules that could guide effective treatment, while others have aimed to predict COVID-19 progression towards MAS [40,41,42,43,44,45].

### 3.2. MAS and Other Viral Infections

#### 3.2.1. Epstein–Barr Virus

Epstein–Barr virus (EBV) is a well-recognized trigger of HLH. HLH can emerge as a result of EBV infection in various clinical contexts, including pHLH in genetically predisposed individuals, sHLH in those without a known genetic predisposition, and HLH associated with EBV-positive neoplasms [46]. Although, EBV-associated HLH is prevalent in East Asia, it has been reported among non-Asian populations, especially in those who are Hispanic [47]. The proposed mechanism by which EBV induces HLH involves the proliferation of cytotoxic T cells and activation of macrophages by EBV-infected B cells, resulting in uncontrolled immune activation and hypercytokinemia. Additionally, EBV targets CD8+ T or NK cells via CD21, leading to uncontrolled cytokine production [48]. A recent study evaluated NK cell cytotoxicity in patients with sHLH and found decreased cytotoxicity among HLH patients. However, no significant difference was observed between patients with EBV-HLH and non-EBV HLH [49]. The diagnosis of HLH in patients with EBV infection can be challenging, as these patients may develop some of the hallmarks of HLH as part of natural infection [50]. The clinical and laboratory characteristics of patients with EBV-HLH, as well as those associated with other viral infections, are presented in Table 1. One study compared the laboratory parameters of patients with EBV and non-EBV sHLH and found that ferritin, LDH, and liver function tests were significantly higher in EBV-associated HLH patients. Also, they reported that EBV DNA load influences disease development in sHLH [51]. Among the diverse etiologies of HLH, EBV-associated HLH has the highest mortality [52]. Thus, it has been suggested that an improved HLH index (IH index), which combines parameters such as sCD25, procalcitonin, and eGFR, can be used to identify adult patients with EBV-HLH who have a poor prognosis. However, further validation will be needed [53]. Importantly, central nervous system (CNS) involvement in EBV-HLH has been associated with poor clinical outcomes [54].

#### 3.2.2. Cytomegalovirus

Other members of the herpes virus family have been associated with HLH. Although CMV-HLH remains poorly characterized, HLH can emerge as a result of primary CMV infection or CMV reactivation [67]. HLH occurs more frequently among immunodeficient individuals; however, it has been reported in several immunocompetent cases [68]. Of note, in immunocompetent patients HLH seems to follow a severe disease course, such as CMV pneumonia [69,70,71]. Regarding immunodeficient cases, an association between CMV-HLH, inflammatory bowel disease (IBD), and immunomodulatory treatment with thiopurines has been described since IBD has been observed as the main comorbidity in this patient population [56,67,72].

#### 3.2.3. Herpes Simplex Virus and Varicella Zoster Virus

Herpes simplex virus (HSV) infection has been rarely associated with HLH. The literature on HSV-HLH is scarce, with only a few case reports available. Most documented cases involve infants with disseminated HSV infection, whereas comparatively fewer cases pertain to adults [73]. Interestingly, even though disseminated HSV infection usually occurs in immunocompromised patients the majority of HSV-HLH, adult patients were immunocompetent [73]. Several authors have noted significantly elevated ferritin levels in patients with HSV-HLH, posing a diagnostic challenge as cytopenia and hyperferritinemia, which characterize HLH, can result solely from disseminated HSV infection [74,75,76,77]. Also, a study conducted in neonates revealed that high ferritin levels, along with other factors, may indicate a more severe form of the disease and are correlated with poor outcomes [78]. Lastly, there are few reported cases of varicella-zoster virus (VZV) infection as a trigger of HLH, primarily in the pediatric population. However, some cases have been associated with underlying immune system disorders, such as adenosine deaminase 2 (ADA2) deficiency or Griscelli syndrome [79,80,81,82].

#### 3.2.4. Influenza Virus

Additionally, MAS is known to complicate influenza virus infection. Severe influenza A (pandemic strain H1N1) infection may develop many similarities with HLH. First, there are plenty of overlapping features in the cytokine storms induced by H1N1 and MAS [13]. Moreover, a study conducted in critically ill patients with H1N1 infection reported similarities in clinical and laboratory parameters among patients who developed HLH secondary to the infection and those who did not. However, HLH patients showed more extensive signs of inflammation, had higher ferritin concentration, and had lower albumin and organomegaly [57]. The same study found an inverse correlation between serum ferritin concentration and NK and cytotoxic T cell percentages which may indicate a potential role for reduced NK cell numbers in the pathogenesis of sHLH [57]. These findings are in line with previous reports indicating the presence of heterozygous mutations in established pHLH genes, which were shown to decrease NK cell cytolytic function among fatal cases of H1N1 infection [83]. Also, HLH is associated with avian influenza (H5N1) infection [84]. In H5N1 infection, excessive IL-18 production and a subsequent cytokine storm result from a prolonged NLRP3 (expressed in macrophages) activation by the PB1F2 viral protein. Thus, IL-18 could potentially contribute to the development of MAS [85].

#### 3.2.5. Human Adenovirus and Parvovirus B19

Human adenovirus infection is seldom associated with HLH. The majority of reported cases pertain to children, while only a few cases have involved adults [58,86,87,88]. A recent study investigated the risk factors associated with HLH development in children with severe adenovirus pneumonia and found that patients with longer duration of fever (over 12.5 days) and elevated triglyceride levels (over 3.02 mmol/L) were at higher risk of developing HLH [89]. Also, parvovirus B19 is another rare HLH trigger, and associated cases seem to follow a relatively benign course compared with other HLH etiologies [90]. Pediatric patients, immunocompromised individuals, and those with hematologic pathology are the populations most commonly affected [60,61,62,91].

#### 3.2.6. Hepatitis Viruses

Moreover, HLH can result from infection with hepatitis viruses, with the majority of cases involving HAV and, less commonly, HBV or HCV infection [92,93,94]. Two systematic reviews of HAV-associated HLH cases have identified hepatomegaly and splenomegaly as distinguishing features of HAV-HLH, as these signs are rarely observed in patients with HAV infection who do not develop HLH [63,95]. Also, they noted that thrombocytopenia could be an early sign of HAV-HLH [63,95]. However, while hyperferritinemia can occur solely due to HAV infection, higher levels may indicate the presence of HLH [63].

#### 3.2.7. Human Immunodeficiency Virus

In HIV-positive patients, HLH can manifest as a complication in various disease-related clinical scenarios. Firstly, HLH may develop during acute HIV infection [64,96,97]. Secondly, it can be triggered by opportunistic infections to which HIV-positive patients are predisposed [98,99,100]. Also, it may arise due to immune reconstitution inflammatory syndrome (IRIS), which is an exaggerated inflammatory immune response following the initiation of highly active antiretroviral treatment (HAART) [101,102]. Lastly, HLH can emerge as a result of virus-associated malignancies [103,104]. The diagnosis of HLH in HIV-positive individuals can be challenging because the clinical and laboratory abnormalities that define HLH may also occur in advanced HIV/AIDS [65]. Maintaining a high degree of clinical suspicion is crucial for diagnosing HLH in HIV-positive patients, particularly those who are non-compliant to ART (uncontrolled HIV), with opportunistic infections. On the other hand, HLH may also occur in patients with low viral loads and CD4+ T cell counts, driven by the HIV-induced immune dysregulation [65].

#### 3.2.8. Hemorrhagic Fevers

Viral hemorrhagic fevers, such as Ebola virus disease and dengue fever, have been associated with HLH. Ebola virus disease, especially severe forms, share significant similarities with HLH. Symptoms such as fever, cytopenia, hyperferritinemia, hypofibrinogenemia, and low NK cell counts that characterize HLH are commonly found in Ebola virus disease. Thus, there has been the hypothesis that infection with Ebola virus is associated with overwhelming macrophage activation [105]. In a recent study, macrophage activation maker soluble CD163 has been found upregulated in severe Ebola virus disease. This finding provides additional support for the hypothesis [106]. Similarly, a hyperinflammatory/MAS phenotype occurs in dengue and is associated with disease severity [107,108,109]. A recent systematic review and meta-analysis noted that dengue-associated HLH has a lower fatality rate compared to other infection-associated HLHs; however, the presence of HLH increases the mortality of dengue patients [66]. Therefore, since the diagnosis of dengue-related HLH could be challenging, clinicians should consider diagnostic indicators such as elevated LDH, ferritin, prolonged fever duration, persistent thrombocytopenia, anemia, and hepatomegaly as potential signs of disease progression towards HLH [66,110].

### 3.3. Therapeutic Options

Concerning the treatment of sHLH/MAS, definitive guidelines have not yet been established. This is partly attributable to the heterogeneity of triggering agents [1]. Management of the condition primarily relies on guidelines and protocols derived from other disease contexts, such as the HLH-2004 protocol. Early recognition and combination chemo-immunotherapy, including corticosteroids, etoposide, and cyclosporine A, constitute the cornerstone of therapeutic options [6]. Additionally, cytokine-targeted therapies which have been previously used to treat autoimmunity-associated MAS are expanding the armamentarium for therapeutic intervention [9]. However, when it comes to infectious HLH, achieving a balance between the immunosuppressive treatment, which is necessary for managing the cytokine storm, and infection control is crucial [56]. Table 2 summarizes the therapeutic regimens that have been used for various viral infections triggering MAS. Although the majority of cases are treated using the standard HLH medications, there is a significant lack of literature regarding studies focusing on the treatment of sHLH, particularly among adult populations.

In a recent study, 93 patients with EBV-associated HLH were divided into two groups according to whether the initial treatment contained etoposide or not. The etoposide group had a better survival rate compared to those who did not receive the drug with significantly higher survival rates among the adult population [135]. Also, two studies evaluated the efficacy of liposomal doxorubicin, etoposide, and methylprednisolone (L-DEP) as an initial therapy for adults with EBV-HLH and as salvage therapy for pediatric patients [136,137]. Additionally, due to EBV’s tropism for B lymphocytes, the anti-CD20 antibody rituximab can be used in combination with other HLH-directed medications for effective infection control. As demonstrated in one study, when combined with conventional HLH therapy, rituximab improves symptoms, reduces viral load, and diminishes inflammation [138]. Recently, a trial investigated the role of ruxolitinib, a Janus kinase (JAK) 1 and 2 inhibitor in the treatment of EBV-associated HLH. In this study, ruxolitinib demonstrated a notably high favorable overall response. Of importance, all the responding patients achieved their first response within 3 days, indicating that ruxolitinib has a rapid onset of action in controlling HLH manifestations [139].

Moreover, the COVID-19 pandemic created an opportunity to conduct larger studies on drugs for the treatment of MAS, particularly focusing on cytokine-targeted therapies. Studies on COVID-19-associated MAS patients showed that early use of IL-1 blocker anakinra reduces the need for intensive care and contributes to the improvement in laboratory and radiological findings [140,141,142]. Another study supported that the addition of anakinra to steroid treatment results in a significant decrease in biochemical parameters. However, no significant difference was observed concerning oxygen requirements or mortality rates between patients treated solely with steroids versus those who received anakinra as well [143]. Regarding IL-6 blockade with tocilizumab in COVID-19-associated MAS, the results are controversial. Some studies have reported that treatment with tocilizumab contributes to reduced mortality and favorable outcomes [144,145]. Conversely, another study noted that tocilizumab demonstrated limited efficacy for COVID-19-related MAS [146]. In another recent trial, tocilizumab led to an increased incidence of secondary infections among critically ill patients with COVID-19-associated MAS, while treatment with anakinra resulted in favorable responses [147]. Lastly, alternative treatment options targeting IL-18, IFN-γ, and the JAK pathway have been under discussion [148].

### 3.4. Conclusions

Although MAS is a rare complication of viral infections, the condition is characterized by poor clinical outcomes. Diagnosing MAS, particularly in adult patients, remains a formidable challenge due to the absence of universally accepted classification and diagnostic criteria. The management of the condition relies on established protocols for pHLH. However, recent advances in cytokine-targeted therapies have demonstrated promising potential, offering hope for improved treatment outcomes.

## Figures and Tables

**Figure 1 viruses-16-01004-f001:**
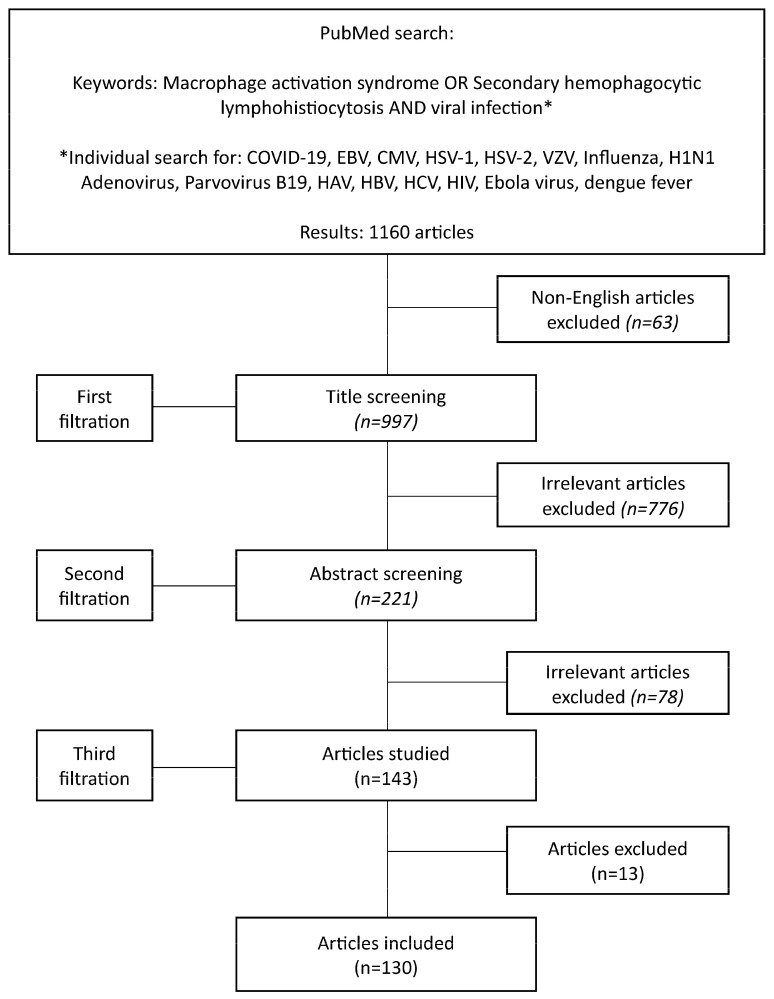
Study flowchart.

**Table 1 viruses-16-01004-t001:** Clinical and laboratory findings of MAS in different viral infections.

Viral Infection	Author	Year	Patient Population	Clinical and Laboratory Findings
**EBV-HLH**	Lai et al. [55]	2018	133 adult patients 91 males 42 females	Fever 91.2% Splenomegaly 89.5% Hyperferritinemia 92% Cytopenia (reduction of 2 or 3 lines) 93.2% Hypertriglyceridemia 34.7% Hypofibrinogenemia 59% Elevated liver enzymes 80% Hemophagocytosis 86.2% sIL-2R 87.7% NK cell activity 44.5%
**CMV-HLH**	Rolsdolph et al. [56]	2022	71 adult patients 29 males 42 females	Fever 71/71 (100%) Hepatomegaly 3/71 Splenomegaly 18/71 Hepatomegaly and splenomegaly 18/71 Hyperferritinemia 61/71 (86.9%) Cytopenia (reduction of at least 2 lines) 57/71 (80.2%) Hypertriglyceridemia 21/71 Hypofibrinogenemia 5/71 Hemophagocytosis 62/71 (87.3%) sIL-2R (results available for 18 patients) 13/18 NK cell activity 6/71
**H1N1-HLH**	Bahr Greenwood et al. [57]	2021	4 adult patients 4 males	Hepatomegaly 4/4 Splenomegaly 3/4 Hyperferritinemia 4/4 Cytopenia (reduction of 2 lines) ¾ Hypertriglyceridemia 4/4 Elevated liver enzymes 4/4 Hemophagocytosis 4/4 sIL-2R (results available for 3 patients) 3/3 NK cell activity 2/4
**Adenovirus-HLH**	Mellon et al. [58]	2016	6 patients 1 adult 5 pediatric 4 males 3 females	Fever 6/6 Hepatomegaly 4/6 Splenomegaly 3/6 Anemia 5/6 Leukopenia 4/6 Thrombocytopenia 5/6 Hypofibrinogenemia 2/6 Elevated liver enzymes 6/6
**Parvovirus B19-HLH**	Leelaviwat et al. [59] Soldo-Juresa et al. [60] Orth et al. [61] Macauley et al. [62]	2023 2010 2022 2019	4 patients 4 females	Fever 4/4 Splenomegaly 2/4 Hyperferritinemia 4/4 Anemia 3/4 Leukopenia 2/4 Thrombopenia 3/4 Hypertriglyceridemia 2/4 Hypofibrinogenemia 2/4 Elevated liver enzymes 3/4 Hemophagocytosis 3/4 sIL-2R 3/4
**HAV-HLH**	Mallick et al. [63]	2019	27 patients 17 adults 10 pediatric 14 males 13 females	Fever 26/27 (96.2%) Hepatomegaly 17/27 (62.9%) Splenomegaly 22/27 (81.4%) Hyperferritinemia (results available for 21 patients) 21/21 (100%) Pancytopenia 15/23 (65.2%) Hypertriglyceridemia (results available for 19 patients) mean value 394.9 (range 74–690) Hypofibrinogenemia (results available for 14 patients) mean value 7.34 (range 0.3–76) Hemophagocytosis 26/27 (96.2%) sIL-2R (results available for 4 patients) 4/4
**HIV-HLH**	Fazal et al. [64]	2020	52 patients 42 males 10 females	Fever 52/52 (100%) Splenomegaly 42/46 (91.3%) Hyperferritinemia 42/43 (97.6%) Anemia 40/52 (76.9%) Leukopenia 22/30 (73.3%) Thrombopenia 47/51 (92.1%) Hypertriglyceridemia 24/33 (72.7%) Hypofibrinogenemia 6/15 (40%) Hemophagocytosis 39/45 (86.6%) sIL-2R 10/11 (90.9%) NK cell activity 2/5 (40%)
	Tabaja et al. [65]	2022	81 patients 63 males 18 females	Fever 80/81 (98.7%) Splenomegaly 52/71 (73.2%) Hyperferritinema 64/64 (100%) Cytopenia 53/70 (75.7%) Hypertriglyceridemia 34/47 (72.3%) Hypofibrinogenemia 10/25 (40%) Hemophagocytosis 70/80 (87.5%) sIL-2R 21/23 (91.3%) NK cell activity 10/10 (100%)
**Dengue-HLH**	Giang et al. [66]	2018	122 patients 62 males 60 females	Fever 97.2% Splenomegaly 78.4% Hepatomegaly 70.2% Hyperferritinemia 97.1% Anemia 76% Thrombopenia 90.1%

This table summarizes the clinical characteristics and laboratory findings of patients that developed MAS in the context of different viral infections. Abbreviations: EBV: Epstein–Barr virus; CMV: cytomegalovirus; HAV: hepatitis A virus; HIV: human immunodeficiency virus; HLH: hemophagocytic lymphohistiocytosis.

**Table 2 viruses-16-01004-t002:** Therapeutic regimens for viral infection associated MAS.

Viral Infection	Reference	Treatment	Outcome	Additional Information
**COVID-19**	Abdelgabar et al., 2022 [111]	antibiotics remdesivir dexamethasone tocilizumab	Died	
	Cruz et al., 2022 [112]	pentaglobin pulse therapy with dexamethasone	Survived	Other regimes did not improve clinical condition
	Hieber et al., 2022 [113]	IVIG dexamethasone anakinra	Survived	MAS due to vaccination anakinra led to improvement
	Reiff et al., 2022 [114]	antibiotics remdesivir dexamethasone IVIG anakinra	Survived	Heterozygous pHLH gene mutation
**EBV**	Macaraeg et al., 2023 [115]	emapalumab acyclovir TMP/SMX	Survived	
	Shaw et al., 2016 [116]	dexamethasone etoposide anti-topoisomerase II agent rituximab	Died	Etoposide discontinued due to worsening liver failure
	Zhu et al., 2023 [117]	PD-1 inhibitor rituximab	Survived	
	Contreras-Chavez et al., 2020 [118]	etoposide dexamethasone rituximab	Died	
	Shakaguchi et al., 2023 [119]	corticosteroids etoposide cyclosporine rituximab	Died	Reduced dose of etoposide due to hepatic dysfunction
	Quadri et al., 2020 [120]	high dose dexamethasone	Died	
	Ioannou et al., 2020 [121]	acyclovir multiple antibiotic courses IVIG prednisolone	Survived	
**CMV**	Awasthi et al., 2020 [122]	ganciclovir high dose steroids IVIG		
	Silwedel et al., 2017 [123]	ganciclovir	Survived	Pediatric case
	Lau et al., 2020 [124]	ganciclovir anakinra dexamethazone etoposide	Survived	Patient did not respond to anakinra and switched to etoposide
	Gullickson et al., 2023 [125]	high dose steroids ganciclovir continuous anakinra infusion	Survived	
**HSV**	McKeone et al., 2021 [126]	dexamethasone etoposide emapalumab anakinra IVIG	Died	Pediatric case
	Mazzotta et al., 2022 [127]	antibiotics acyclovir dexamethasone	Died	
**Influenza**	Jayashree et al., 2017 [128]	methylprednisolone antibiotics IVIG	Survived	Pediatric case MAS due to influenza B virus
	Casciaro et al., 2014 [129]	prednisolone antibiotics oseltamivir	Survived	Pediatric case
**Parvovirus B19**	Leelaviwat et al., 2023 [59]	high-dose methylprednisolone etoposide	Died	
	Macauley et al., 2019 [62]	etoposide high-dose dexamethasone IVIG	Survived	
**HAV**	Dogan et al., 2021 [130]	IVIG dexamethasone	Survived	
**HIV**	Tong et al., 2019 [131]	etoposide dexamethazone	Died	
	Kim et al., 2021 [96]	high-dose dexamethasone elvitegravir/cobicistat/emtricitabine/tenofovir	Survived	
**Dengue**	Ray et al., 2017 [132]	IVIG	Survived	
	Acharya et al., 2022 [133]	antibiotics dexamethasone etoposide	Survived	
	Pradeep et al., 2023 [134]	dexamethasone	Survived	

This table summarizes the therapeutic regimens that have been used in reported cases of MAS triggered by viral infections. Abbreviations: MAS: macrophage activation syndrome; COVID-19: corona virus disease 2019; EBV: Epstein–Barr virus; CMV: cytomegalovirus; HSV: herpes simplex virus; HAV: hepatitis A virus; HIV: human immunodeficiency virus; PD-1: programmed cell death protein 1; IVIG: intravenous immune globulin.

## Data Availability

No new data were created or analyzed in this study. Data sharing is not applicable to this article.

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
