# Peer review of "Macrophage Activation Syndrome in Viral Sepsis"

_viruses, 2024, doi:10.3390/v16071004_

Round 1

Reviewer 1 Report

Comments and Suggestions for Authors

In this review of macrophage activation syndrome (MAS) in viral sepsis, Papageorgiou et al conduct a literature review of articles covering MAS and specific viral infections. This is a well written review with informative tables that summarize the literature. It will likely be a welcome addition to the literature.

A couple of comments to improve the review are below.  

1)      In the methods section, more detail could be provided on who conducted the literature review. Did all the authors read all the articles. Did more than one author have to agree on excluding an article?  

2)      Some strange citations are included in the introduction. For example, #9 seems to cite an abstract instead of the published manuscript describing the 2016 Classification Criteria for MAS in systemic JIA.

3)      In the COVID19 section, there are multiple statements that patients with COVID19 in the ICU have MAS. For example, “COVID19 causes changes in cytokine profiles and these changes are consistent with the presence of MAS.” This is a very tricky topic. There is no agreed upon definition for MAS aside for patients with systemic JIA. And it does not appear that any of the described papers reviewed by the author used established HLH/MAS definitions to confirm that patients with COVID19 in the ICU had MAS. So, it would be better if this language was softened and terms like resembled MAS or was similar to MAS.

4)      In the COVID19 section, the authors could consider touching on MIS-C, which has an IFNg signature like MAS and some articles have even applied MAS/HLH criteria to patients with MIS-C.

5)      The hemorrhagic fevers aren’t included in Table 1.

6)      There is a large cohort of patients with EBV HLH from China treated with ruxolitinib that should be added to the treatment section. PMID: 35344583. I didn’t see this cited in the manuscript.

7)      Some type of summary or conclusion section might be useful.

Reviewer 2 Report

Comments and Suggestions for Authors

This review summarizes current evidence on the etiology and therapy of macrophage activation syndrome (MAS). The manuscript is well-written and clear. I read it with great interest. Here are a few comments for your consideration:

1.     The authors mentioned that MAS is a form of secondary hemophagocytic lymphohistiocytosis (HLH). However, distinguishing between secondary HLH and primary HLH in clinical settings is challenging. MAS and HLH are sometimes used interchangeably or misused in studies. It is important to address this issue in the introduction. Including a figure or table to illustrate the spectrum of these terms would enhance clarity for readers.

2.     The layout of Table 1 is difficult to read. Please consider revising it for better readability before publication.

3.     The aim of this article is not to review all manifestations of MAS across different viral etiologies, which results in some studies being excluded from Table 1. Providing a rationale or explanation for the selection of the cited studies would be helpful. For instance, explaining the choice of the largest retrospective study or the latest systematic review would add context.

4.     There are a few capitalization errors that need correction. For example, "Anakinra" in Table 2 should be "anakinra."

Comments on the Quality of English Language

The quality is good.
